# Management of Cancer-Associated Thrombosis in France: A National Survey among Vascular Disease and Supportive Care Specialists

**DOI:** 10.3390/cancers14174143

**Published:** 2022-08-27

**Authors:** Isabelle Mahé, Céline Chapelle, Ludovic Plaisance, Laurent Bertoletti, Patrick Mismetti, Didier Mayeur, Guillaume Mahé, Francis Couturaud

**Affiliations:** 1Université Paris Cité, 75006 Paris, France; 2Service de Médecine Interne, Hôpital Louis Mourier, AP-HP, 92700 Colombes, France; 3Innovative Therapies in Haemostasis, INSERM UMR_S1140, 75006 Paris, France; 4INNOVTE-FCRIN (Investigation Network On Venous Thrombo-Embolism), 42055 Saint-Etienne, France; 5Unité de Recherche Clinique Innovation et Pharmacologie, CHU de Saint-Etienne, 42270 Saint-Etienne, France; 6SAINBIOSE INSERM U1059, Université Jean Monnet, Université de Lyon, 42023 Saint-Etienne, France; 7Service de Médecine Vasculaire et Thérapeutique, CHU de St-Etienne, 42055 Saint-Etienne, France; 8INSERM CIC-1408, CHU de St-Etienne, 42055 Saint-Etienne, France; 9Oncology Department, Centre Georges François Leclerc, 1, Rue Professeur Marion, BP77980, CEDEX, 21079 Dijon, France; 10Unité de Médecine Vasculaire, CHU Rennes, 35000 Rennes, France; 11INSERM CIC 1414, 35200 Rennes, France; 12University of Rennes 1, 35000 Rennes, France; 13University of Rennes 2, M2S-EA 7470, 35000 Rennes, France; 14Département de Médecine Interne et Pneumologie, INSERM U1304 (GETBO), CHU Brest, Université de Bretagne Occidentale, 29238 Brest, France

**Keywords:** cancer, venous thromboembolism, anticoagulants, treatment guidelines, survey

## Abstract

**Simple Summary:**

Patients with venous thromboembolism events in the context of cancer should receive anticoagulants for at least 6 months. Both low molecular weight heparins (LMWHs) and direct oral anticoagulants (DOACs) are considered in international guidelines, with a different approach. To investigate the determinants of decision, at thrombosis diagnosis and after 6 months, and the practices when facing special situations, such as venous thromboembolic recurrence or thrombocytopenia, we designed a survey among specialists of cancer-associated thromboembolism, including vignettes about patients with different cancer sites and questions. We considered points related to cancer disease, anticancer treatments and characteristics of patients.

**Abstract:**

Low molecular weight heparins (LMWHs) are recommended by international guidelines for at least 6 months in patients with cancer-associated thromboembolism (CAT). Direct oral anticoagulants (DOACs) have been proposed as an alternative to LMWH. In clinical practice, the specialists in charge of CAT have to decide which anticoagulant to prescribe. An electronic survey tool, including vignettes and questions, was sent to members of the French Society of Vascular Medicine, the French-speaking association for supportive care in oncology and the Investigation Network On Venous Thrombo-Embolism. Among the 376 respondents, LMWHs were reported as the first choice by most specialists. The prescription of DOACs within the first 3 weeks of CAT diagnosis was highly dependent on the cancer site: 5.9%, 18.6% and 24.5% in patients with locally advanced colorectal, lung and breast cancer, respectively. The determinants were mostly related to cancer (site and stage or evolution) and to anticancer treatments. For 61% of physicians, some anticancer treatments were contraindications to DOACs. However, almost 90% of physicians considered switching to DOAC after a median 3-month period of LMWHs. In daily practice, LMWHs and DOACs are now considered by specialists of CAT; the decision is mostly driven by the site of cancer. The role of anticancer treatments in the decision remains to be investigated.

## 1. Introduction

Patients with cancer are at a significantly higher risk of developing venous thromboembolism (VTE), including deep vein thrombosis (DVT) and pulmonary embolism (PE), than the general population. According to various studies, VTE represents one of the most important causes of morbidity in cancer patients, affecting up to 20% of patients with cancer, and the occurrence of VTE is associated with increased mortality [1]. Of note, the incidence of cancer-associated thrombosis (CAT) has increased over time [1,2].

For many years, low molecular weight heparins (LMWHs) have been recommended by international guidelines as a first-line anticoagulant therapy owing to their superiority to vitamin K antagonists (VKA) in preventing VTE recurrence [3]; they should be administered for at least 6 months in patients with CAT. Randomised controlled trials have established the non-inferiority of direct oral anticoagulants (DOACs) as compared to LMWHs in CAT treatment and even a superiority on the risk of VTE recurrences in the meta-analysis (HR = 0.63; 95% CI: 0.47–0.86) [4]. The risk of major bleeding on DOAC as compared to dalteparin is different according to the study (HR = 1.74; 95% CI: 0.95–3.18) [5] and in the SELECT D Study (HR = 1.83; 95% CI: 0.68–4.96) [6] but not in the CARAVAGGIO study (HR = 0.82; 95% CI: 0.40–1.69) [7], and overall, the risk of clinically relevant bleedings (sum of major bleeding and clinically relevant non-major bleedings) was increased in DOACs as compared to LMWHs (HR = 1.47; 95% CI: 1.17–1.84) [4]. DOACs have therefore been proposed as a treatment option by many guidelines for the initial treatment of patients with CAT for up to 6 months [8,9,10,11].

However, there are remaining unanswered questions. In particular, the risk of gastrointestinal (GI) bleeding is a concern with DOACs, specifically for patients with gastrointestinal cancer [6,12]. Additionally, there is increasing questioning of the potential consequences related to the concomitant prescription of DOACs and some anticancer treatments and, more importantly, their clinical relevance [13,14,15,16]). The occurrence of complications, such as VTE recurrence or thrombocytopenia, frequently complicates the management of patients with CAT, with few available data in the literature to help clinicians facing those situations [17].

In clinical practice, and according to the recent clinical guidelines, the specialists in charge of CAT management have to decide which anticoagulant to prescribe when facing a patient with CAT, but less is known about the reasons influencing the anticoagulant treatment strategy in real-world practice.

In order to describe the therapeutic management proposed for patients diagnosed with CAT, we therefore designed a survey with different clinical scenarios to assess the clinicians’ reported treatment preferences according to the clinical setting.

## 2. Materials and Methods

### 2.1. Survey Objective

The objective of this survey was to describe the therapeutic management proposed for patients diagnosed with CAT and to identify the factors influencing the therapeutic decision according to vascular and oncology specialists in managing CAT patients in France.

### 2.2. Survey Design

A French online questionnaire, including vignettes and questions about the management of patients with CAT, was developed by a panel of health professionals from different disciplines, including vascular medicine, oncology, oncologic supportive care. This survey was performed under the coordination of the Investigation Network On Venous Thrombo-Embolism (INNOVTE) and was supported by the French Society of Vascular Medicine (SFMV) and the French-speaking association for supportive care in oncology (AFSOS). Bayer provided an unrestricted educational grant for this study.

The survey was designed in order to reflect as much as possible the clinical situations encountered in everyday practice. Accordingly, the 16-question survey consisted of: (1) several short questions regarding the participants’: medical speciality (vascular medicine, cardiology, pneumology, internal medicine, oncology, radiotherapy, supportive care, other), type (public, private, private health establishment of collective interest) and department of practice and age group (less than 40 years, 40–49 years, 50–59 years, 60 years and more); (2) questions regarding the clinical practice; (3) vignettes about treatment management based on several clinical cases, i.e., patients with symptomatic proximal DVT and PE and an advanced cancer with different sites (lung, colorectal, breast); (4) vignettes about the factors influencing the management of patients with CAT, including cancer site, cancer treatment, thrombosis site; (5) vignettes about the management of special situations occurring in patients on anticoagulant treatment (VTE recurrence, thrombocytopenia); (6) questions about the recommendations used in practice and clinical studies that have had an impact on patient management. All vignettes were single-choice responses. Questions were mapped to clinical themes and analysed to uncover physician gaps by speciality.

### 2.3. Survey Population and Survey Administration

The members of the associations at the time of the survey were invited by email to participate in the online electronic survey. Survey participation was voluntary and anonymous, and the responses were collected electronically using Google Forms Survey^®^ to assess the opinions of the experts because it is easily and freely accessible. The survey was first tested for clarity and content among the members of the steering committee before being sent to members of the three associations (INNOVTE, SFMV, AFSOS).

### 2.4. Data Analysis

The analysis of the collected information was only descriptive; no comparison tests were performed. The results were presented according to the medical speciality: cancer-related or vascular-related specialists. The responses of the survey were reported as means [± standard deviation (SD)], median and interquartile range (IQR) for continuous data, and number and percentages for categorical data. The percentages were presented with a corresponding 95% confidence interval (95% CI).

Statistical analyses were performed using SAS version 9.4 software (SAS Institute Inc, Cary, NC, USA).

## 3. Results

Between 9 September 2020 and 7 January 2021, among the 2104 members of the associations invited, 414 specialists responded to the survey (response rate 20%), of whom 376 were in charge of patients with CAT and were included in the analysis: 40 cancer-related specialists (10.6%), 336 vascular-related specialists (89.4%). Cancer-related specialities were oncology (47.5%), supportive care (42.5%) and others (10.0%), and clinicians in vascular diseases care were vascular medicine (88.4%), internal medicine (5.3%) and respiratory disease specialists (2.4%), cardiologists (1.8%) and others specialities (2.1%).

The participants’ characteristics are summarised in Table 1. The majority of participants were older than 50 years (56.1%). The mode of practice was mainly in a public hospital for cancer-related specialists (55.0%) and in a private hospital for vascular-related specialists (63.4%).

On average, respectively, 51.3% and 37.3% of cancer-related and vascular-related specialists managed more than five patients with CAT per month for the initial treatment (3 to 6 months) and 75.7% and 39.0%, respectively, for the prolonged treatment beyond 6 months (Figure 1).

### 3.1. Management of a CAT, Choice of Anticoagulant Therapy and Tumour Site

For the initial CAT treatment in the context of advanced cancer (lung, colorectal, breast), LMWHs were the first-choice treatment for most specialists (92.8%, [95% CI 90.2–95.4%]); in particular, over 90% of specialists (n = 345, 91.8% [95% CI 89.0–94.5%]) prescribed LMWHs in the case of colorectal cancer (Figure 2). DOACs were less often preferred in patients with colorectal cancer than in patients with lung or breast cancer (5.9% [95% CI 3.5–8.2%], 18.6% [95% CI 14.7–22.5%] and 24.5% [95% CI 20.1–28.8%], respectively). The same initial therapeutic option, irrespective of the primary tumour’s site, was chosen by 76.9% (95% CI 72.6–81.1%) of specialists (almost 95% (95% CI 92.2–97.4%) of specialists selected LMWHs, and DOACs were selected by 3.5% (95% CI 1.3–5.6%) of specialists).

### 3.2. CAT Treatment, Reasons for the Choice

#### 3.2.1. Initial Treatment

The reported determinants driving the decision about the initial anticoagulant treatment of CAT according to the medical speciality are summarised in Table 2. The most frequently reported reasons driving the treatment decision were the stage and/or evolution of the cancer (39.6% [95% CI 34.7–44.6%]), the site of cancer (37.0% [95% CI 32.1–41.8%]), the patients’ comorbidities (33.0% [28.2–37.7%]) and the risk of drug interaction (31.6% [95% CI 26.9–36.3%]). The advocated reasons were very similar across cancer-related and vascular-related specialists. Most specialists considered that the presence of an ongoing anticancer treatment is a contraindication to DOACs (61.2% [95% CI 56.2–66.1%]), mostly because of a perceived risk of bleeding (61.3% [95% CI 55.0–67.6%]) or thromboembolism (23.0% [95% CI 17.6–28.5%]).

#### 3.2.2. After Initial Treatment with LMWHs: Switch to DOACs?

Overall, for patients initiated with LMWH, 88.8% [95% CI 85.6–92.0%] of physicians might consider a switch to DOACs after a median minimal period of 3 months of LMWH, the decision being based on the site of cancer (40.1% [95% CI 34.9–45.4%]), the stage and/or evolution of cancer (38.9% [95% CI 33.7–44.1%]) and patient preference (38.3% [95% CI 33.1–43.5%]) (Table 2).

### 3.3. Results: CAT and Specific Situations

Management of patients with thrombocytopenia at Day 22 from anticoagulant treatment initiation.

The options considered by the respondents when facing a patient undergoing thrombocytopenia during the course of anticoagulant treatment for CAT management are presented in Table 3.

For patients with a platelet count of less than 50 G/L, the practices vary widely; up to 19% (95% CI 15.2–23.1%) of clinicians continue the ongoing treatment with DOAC vs. only 8% (95% CI 5.2–10.7%) with LMWH. Of note, no clinician considered platelet transfusion or inferior vena cava (IVC) filter insertion in the case of a platelet count of less than 50G/l with LMWH, while 13% (95% CI 9.6–16.4%) would proceed to platelet transfusion and 6.6% (95% CI 4.1–9.2%) to IVC filter insertion in the case of treatment with DOAC.

For patients with a platelet count of less than 50 G/l and treated with LMWH, a dosage reduction would be considered by 40.4% (95% CI 35.5–45.4%) of clinicians: LMWH dose reduction by 22.9% (95% CI 18.6–27.1%), switch to a prophylactic dose of LMWH by 12.2% (95% CI 8.9–15.5%) and switch to a reduced dose of DOAC by 5.3% (95% CI 3.0–7.6%) (Table 3).

For patients with a platelet count of less than 50 G/L under DOAC, a dosage reduction would be considered by 32.2% (95% CI 27.6–36.9%) of clinicians as follows: DOAC dose reduction by 10.4% (95% CI 7.3–13.4%), switch to a prophylactic dose of LMWH by 8.8% (95% CI 5.9–11.6%), switch to a 50% reduced dose of LMWH by 13.0% (95% CI 9.6–16.4%).

Of note, a substantial proportion of clinicians considered fondaparinux an option in the case of severe thrombocytopenia (18.6% [95% CI 14.7–22.5%] in the case of LMWH therapy and 10.1% [95% CI 7.1–13.1%] in the case of the DOAC treatment).

In the case of platelet count < 75 G/L, most clinicians would continue the full-dose anticoagulant treatment (with LMWH in 56% [95% CI 51.1–61.1%], DOAC in 22.6% [95% CI 18.4–26.8%] or fondaparinux in 10% [95% CI 7.1–13.1%]).

#### Management of Patients with VTE Recurrence during Anticoagulant Treatment

The options considered by physicians when facing a patient undergoing VTE recurrence during the course of anticoagulant treatment for CAT management (after ruling out a HIT) are presented in Table 4.

In the case of VTE recurrence occurring during LMWH therapy, a large majority of clinicians would use an increased dose of LMWH (76.1% [95% CI 71.7–80.4%]); the other options comprised a switch to DOAC and IVC filter insertion.

In the case of VTE recurrence occurring during DOAC therapy, most clinicians would move to LMWH (91.5% [95% CI 88.7–94.3%]), either at a curative dose (82.2% [95% CI 78.3–86.0%]) or at increased dosage (9.3% [95% CI 6.4–12.2%]).

### 3.4. Guidelines Followed by Practitioners and Studies That Have Influenced Practices

The guidelines considered for practice in CAT management are presented in Figure 3A. This includes mostly French guidelines (French Intersociety guidelines (released in 2019 and 2021) 50.0%, SFMV (released in 2019) 44.7%, AFSOS (released in 2019) 17.0%) and, to a lesser extent, international guidelines (European Society of Cardiology 20.7%, International Initiative on Thrombosis and Cancer 8.2%, International Society Thrombosis and Haemostasis 7.7%, American Society of Clinical Oncology 12.5%). Specialists in vascular disease adhered more to guidelines from the thrombosis groups and specialists in cancer disease to guidelines from the cancer groups. Specialists were influenced by the results of clinical trials that they were aware of (Figure 3B); in particular, 37.5% of cancer-related specialists and 50.9% of vascular-related specialists were influenced by the CARAVAGGIO study [7].

CAT: cancer-associated thrombosis; AFSOS: Association Francophone pour les Soins Oncologiques de Support; ASCO: American Society of Clinical Oncology; ESC: European Society of Cardiology; INNOVTE: Investigation Network On Venous Thrombo-Embolism; ISTH: International Society on Thrombosis and Haemostasis; ITAC: International Initiative on Thrombosis and Cancer; SFMV: Société Française de Médecine Vasculaire.

## 4. Discussion

In this survey of 376 cancer-related and vascular-related specialists regarding their treatment preference for the management of CAT patients, both DOACs and LMWHs were considered by the participants. In the case of overtly symptomatic recent VTE in a patient with an active cancer, the treatment options were consistent with the ongoing guidelines, since 98% of clinicians chose long-term LMWHs or DOACS within the first 3 weeks as options. Only 2% decided to treat patients with VKA. Our results are in line with the survey we performed 4 years ago [18], where more than 90% of respondents used long-term LMWH in accordance with the guidelines that were ongoing at the time of the survey. Overall, it appears that there has been an evolution in the management of patients with CAT; even if the majority of the responding specialists still initiate CAT patients with LMWH, LMWH is not the only option considered by clinicians anymore. Indeed, we observed that only a few months after the publication of the phase-III trials resulting in the possibility offered by the guidelines to opt for DOACs or LMWH as the first option for the initial treatment of patients with CAT, DOACs appeared as an attractive option, either upon VTE diagnosis (20%) or after a median period of 3 months (88%). This reflects that clinicians have accepted DOACs as a relevant option for treating patients with CAT.

Data regarding the impact of new data coming from recent clinical trials and guidelines on the prescription patterns in clinical practice are very sparse and in accordance with our results. In a single-centre retrospective experience from 2016 to 2019 [19], the prescribing patterns in 221 patients with CAT were reported and compared before/after the publication of the Hokusai-Cancer study. LMWHs only were mostly preferred at treatment initiation (80% vs. 4% for DOACs only), and 14.5% had an anticoagulant class change (LMWH to DOAC; 78.1%) at a median of 25 days (IQR 16–30). In the chronic phase, the anticoagulation therapy for patients consisted of LMWH only for 35.8%; there was an increase in the number of patients on DOAC only relative to the acute phase (11.3% vs. 4.1%) and in those experiencing an anticoagulant class change (42.9% vs. 14.5%). Changing from a LMWH to a DOAC was most common (90.1%) and occurred at a median of 121 days (interquartile range [IQR] 110–191). Practices have evolved, since the use of DOACs only in the acute and chronic phases prior to the Hokusai-Cancer trial was 1.0% and 2.0%, respectively, and following publication, it was 6.8% and 19.6%.

In the prospective non-interventional, single-arm cohort COSIMO study [20], patients receiving standard anticoagulation therapy (LMWH or VKA) for ≥4 weeks who were switched to rivaroxaban at the discretion of the treating physician were included. Interestingly, in line with our results, the median duration of all anticoagulant treatments before switching to rivaroxaban was 100 days (IQR, 47–181 days). The most common reasons for changing to rivaroxaban were patient preference factors, including the desire to cease parenteral administration (n = 136; 26.9%), improve the quality of life (n = 94; 18.6%), patient decision (n = 76; 15.0%) and an undesirably long distance from their physician (n = 4; 0.8%), as well as physician decision (n = 174; 34.5%). The identification of the treatment that is most suitable for each patient with CAT remains to be determined.

In our survey, the decision was mainly driven by the cancer situation. Indeed, the most frequently reported determinants driving the treatment decision in favour of the initial anticoagulant treatment of CAT were the stage and/or evolution of cancer, the site of cancer, patient comorbidities and the risk of drug interaction. Moreover, the site of cancer was identified as crucial for the decision at different times of CAT management. For decision at the time of VTE diagnosis, we observed a lower rate of clinicians opting forDOACs therapy in patients with colorectal cancer than for other sites (lung and breast). In addition, the site of cancer was considered an important decision determinant by up to 37% of respondents, and finally, the most important in deciding on the switch to DOACs for patients started on LMWH (up to 40% of specialists). Additionally, for CAT patients treated with DOACs, the type of cancer raised concerns about the bleeding risk among 38% of specialists.

This reflects the questioning regarding the safety of DOACs in patients with gastrointestinal cancer. In the SELECT D Study [6], the recruitment was prematurely interrupted in patients with esogastric cancer due to an excess of major bleeding. A post hoc analysis of the HOKUSAI Cancer Study showed that patients with gastrointestinal cancer were those who underwent an excess of major bleeding when receiving edoxaban as compared with dalteparin [9,12]. Only half of the major bleeding cases were related to the site of cancer. In those trials, a post hoc analysis showed that there was an excess of major bleeding in patients with gastrointestinal cancers receiving DOACs as compared with those on LMWHs (HR = 2.55; 95% CI: 1.24–5.27), while there was no difference in patients with other sites of cancers (HR = 0.80; 95% CI: 0.36–1.77). No difference of major bleeding was observed in the CARAVAGGIO Study between the apixaban and the dalteparin groups (HR = 0.82; 95% CI: 0.40–1.69), also in patients with gastrointestinal cancer [21,22]. The guidelines took those observations differently into account [8,9,10,11].

Otherwise, the concomitant prescription of DOACs and anticancer treatments is an important issue for clinicians. They are afraid of the risk of both bleeding and thromboembolic events in relation to the interactions. Even if no clinical relevance has been demonstrated yet, they are worried about the possible interaction (mostly for drugs with both CYP 3A4 and pGP metabolism), while an alternative (LMWH) at least as effective is available but requires injections. This issue is very complex, since not only the pharmacodynamic interaction but also the characteristics of patients (absorption, renal and hepatic impairment…) could interfere with the level of interaction and the clinical impact [13,14,16]. Studies with different patient profiles and clinical outcomes are definitely required on this matter.

There remain unaddressed needs in the daily management of patients with CAT; this is the case of patients with thrombocytopenia occurrence or patients who undergo VTE recurrence during anticoagulant treatment. The guidelines propose algorithms with a low level of recommendation for managing those situations due to the paucity of data [8,9,10,11,17].

Thrombocytopenia is a frequently encountered situation in patients with cancer, mostly due to anticancer toxicities. The proposed clinical situation took place during the first month following the index event, i.e., the period where the patient is at the highest risk of VTE recurrence. This patient with CAT treated with an anticoagulant is, therefore, at a high risk of VTE recurrence and bleeding. In this situation, both the ongoing drug therapy and the platelet count appear important for making a decision. Out of the proposed options, almost 40% of clinicians reduced the dose in the case of severe thrombocytopenia and a platelet count of less than 50,000 vs. only 8.5% in the case of a platelet count of less than 75,000, reflecting the different perceived risk of bleeding in this case. Surprisingly, fondaparinux was a frequently reported option; the risk/suspicion of heparin-induced thrombocytopenia does not appear to be the explanation, since this was observed for patients on LMWH or DOAC. However, one could assume that the risk of bleeding on a full-dose LMWH or DOAC therapy is similar to that on fondaparinux.

For patients with VTE recurrence on an anticoagulant therapy, a few observational data are available for patients on LMWH [23,24], suggesting that increasing the dosage of LMWH to a full dose in patients on reduced dosage and to increased dosage (+25%) in patients on a full dose is effective and safe. Logically, the switch to LMWH is the preferred option in patients on DOAC at the time of recurrence. Those options are currently being assessed in a prospective REDUCE Study (NCT05229471). All responses of the clinicians are in line with those proposals.

Overall, clinicians appeared to be very well informed on the recently published studies and guidelines. The treatments proposed by clinicians for the initial CAT treatment were in accordance with the guidelines in our two surveys (2017 and 2021) [18]. For clinical practice, the influence of French guidelines (both released by supportive care specialists or vascular specialists) looks the most substantial. This is an important fact to know for devising the actions and ways of disseminating the information to colleagues, especially about a transversal disease that could be managed by different specialities, considering the course and the journey of a patient with a cancer disease. The timing of the guidelines’ release is delayed compared to the communication of the results of clinical trials. It is important to stress that guidelines are not the only factor influencing the practices; at the time of our survey, the ongoing guidelines were in favour of LMWH as a first therapy and suggested the available DOACs in the absence of gastrointestinal or urologic cancer. Respondents were also influenced by the results of clinical trials they were aware of; the practices of 49.5% of respondents were influenced by the results of the CARAVAGGIO study [7], while only 22% and 17% were influenced by the results of the HOKUSAI cancer [5] and SELECT D studies [6].

Thus, different conclusions were drawn from an American survey [25]; physicians had poor knowledge of LWMH and DOAC clinical trial data in CAT, the risk of bleeding and the current guideline recommendations for managing CAT. The majority also had difficulty in selecting the appropriate anticoagulant treatment for patients with CAT. On average, only 21% of participants felt confident in selecting the appropriate anticoagulant strategy for patients with CAT.

The role of medical societies in promoting the results of clinical trials and guidelines is of major importance for improving both the knowledge and the implementation of recommendations.

There are several limitations to our survey. First of all, the response rate was 20%. Second, the responses to the questionnaires were declarative and not based on prescriptions. We attempted to be as close as possible to the practice by proposing simple vignettes and a single response possibility. Third, we had to make a selection of the situations to incorporate into the survey to make it feasible; we must acknowledge that other important situations in patients with CAT were not raised in the present study, such as central catheter thrombosis or renal insufficiency, which may be the subject of another survey. The strength of the study is in proposing an approach to practices in a single survey, including treatment choices and reasons for the preferences.

## 5. Conclusions

Our survey showed that, in daily practice, LMWHs and DOACs are both now considered by specialists of CAT and that the decision is mostly driven by the site of cancer. The evolution and appropriate management of the anticoagulant treatment in patients with symptomatic CAT will continue as long as the clinical practice guidelines are clear and homogeneous, supported by clinical trials. Therapeutic strategies are empiric in situations with an absence of clear guidance or in the case of controversies. This outlines the need for continuing to update the clinical practice guidelines and further information and education of the health professionals managing patients with CAT.

## Figures and Tables

**Figure 1 cancers-14-04143-f001:**
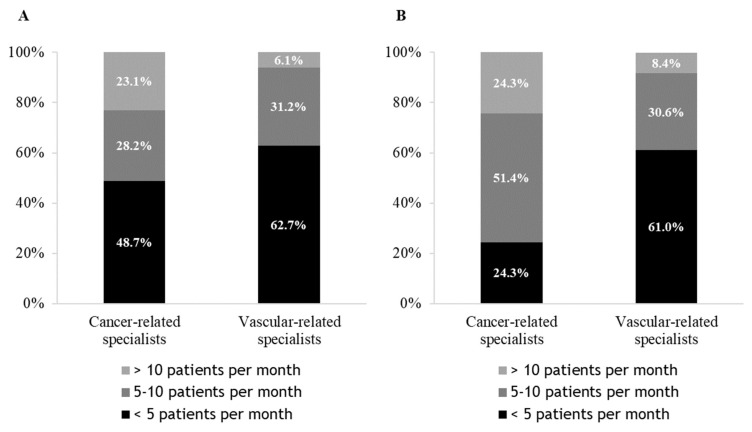
Prescription of curative treatment for cancer-associated venous thromboembolism during (**A**) the initial phase (3 to 6 months); (**B**) the prolonged phase (>6 months).

**Figure 2 cancers-14-04143-f002:**
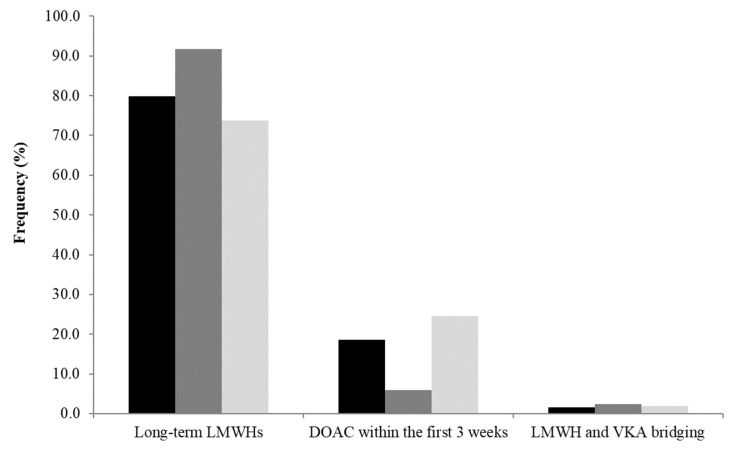
Anticoagulants initiated in a patient with a proximal thromboembolic event and an advanced cancer (according to the site: lung, colorectal, breast). Black: adenocarcinoma of the lung; Dark grey: colorectal cancer; Light grey: breast cancer. DOAC: Direct oral anticoagulant; LMWH: low molecular weight heparin; VKA: vitamin K antagonist.

**Figure 3 cancers-14-04143-f003:**
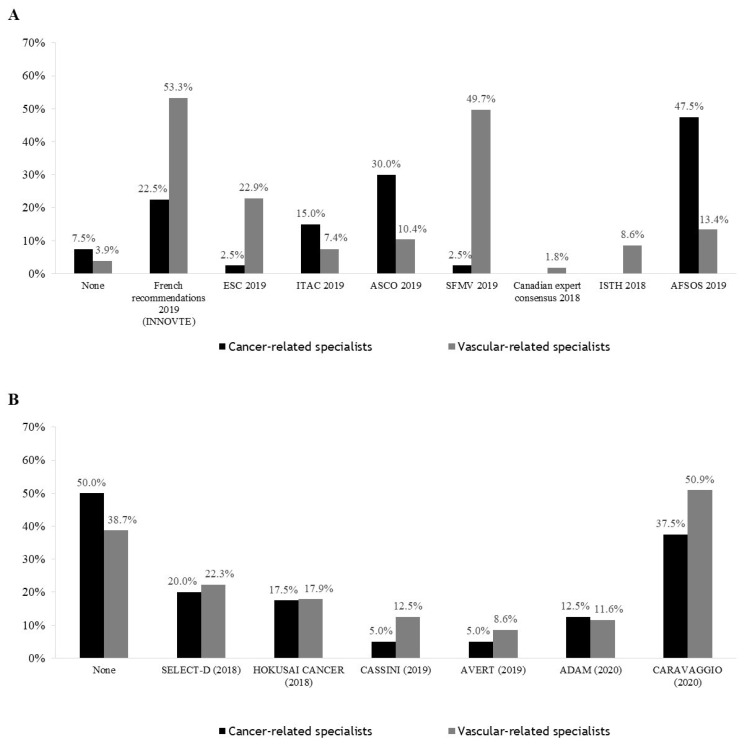
Comparisons of guidelines considered for practice in CAT management (**A**) and studies that have influenced practices (**B**) between cancer-related and vascular-related specialists.

**Table 1 cancers-14-04143-t001:** Participants’ characteristics according to the medical speciality.

	Cancer-Related Specialists (N = 40)	Vascular-Related Specialists (N = 336)	Total (N = 376)
**Age**			
Less than 40 years	23 (57.5%)	66 (19.6%)	89 (23.7%)
40–49 years	7 (17.5%)	69 (20.5%)	76 (20.2%)
50–59 years	4 (10.0%)	118 (35.1%)	122 (32.4%)
60 years and more	6 (15.0%)	83 (24.7%)	89 (23.7%)
**Mode of practice**			
Public hospital	22 (55.0%)	123 (36.6%)	145 (38.6%)
Private	18 (45.0%)	213 (63.4%)	231 (61.4%)

**Table 2 cancers-14-04143-t002:** Determinants driving the treatment decision of the initial anticoagulant treatment of CAT; and decision to switch to DOACs in patients started on LMWHs, according to the medical speciality.

	Initial Treatment			Consider Switching to DOAC * (N = 376)
	Cancer Specialists (N = 40)	Vascular Specialists (N = 336)	Total (N = 376)	
	-	-	-	334 (88.8% [85.6–92.0])
Stage and/or evolution of the cancer	19 (47.5% [32.0–63.0])	130 (38.7% [33.5–43.9])	149 (39.6% [34.7–44.6])	130 (38.9% [33.7–44.1])
Site of cancer	13 (32.5% [18.0–47.0])	126 (37.5% [32.3–42.7])	139 (37.0% [32.1–41.8])	134 (40.1% [34.9–45.4])
Patient comorbidities/additional risk factors	15 (37.5% [22.5–52.5])	109 (32.4% [27.4–37.4])	124 (33.0% [28.2–37.7])	91 (27.2% [22.5–32.0])
Risk of drug interaction	16 (40.0% [24.8–55.2])	103 (30.7% [25.7–35.6])	119 (31.6% [26.9–36.3])	75 (22.5% [18.0–26.9])
Anticancer treatment	8 (20.0% [7.6–32.4])	56 (16.7% [12.7–20.6])	64 (17.0% [13.2–20.8])	57 (17.1% [13.0–21.1])
Type of index event	2 (5.0% [0.0–11.7])	55 (16.4% [12.4–20.3])	57 (15.2% [11.5–18.8])	23 (6.9% [4.2–9.6])
Patient preferences	0 (0.0%)	40 (11.9% [8.4–15.4])	40 (10.6% [7.5–13.7])	128 (38.3% [33.1–43.5])
**Some anticancer treatments are** **contraindications to DOACs**	26 (65.0% [50.2–79.8])	204 (60.7% [55.5–65.9])	230 (61.2% [56.2–66.1])	
**Main reason**				
Haemorrhagic risk	21 (80.8% [65.6–95.9])	120 (58.8% [52.1–65.6])	141 (61.3% [55.0–67.6])	
Thromboembolic risk	0 (0.0%)	53 (26.0% [20.0–32.0])	53 (23.0% [17.6–28.5])	
Toxicity of antitumour treatment	2 (7.7% [0.0–17.9])	23 (11.3% [6.9–15.6])	25 (10.9% [6.8–14.9])	
Inefficacy of anticancer treatment	3 (11.5% [0.0–23.8])	8 (3.9% [1.3–6.6])	11 (4.8% [2.0–7.5])	

The values in square brackets correspond to the 95% confidence interval of the proportion. CAT: cancer-associated thrombosis; DOAC: Direct oral anticoagulant. * from the initial treatment by LMWH in the absence of recurrent or haemorrhagic events.

**Table 3 cancers-14-04143-t003:** First-line proposed management of patients with thrombocytopenia at Day 22 of anticoagulant treatment.

	**LMWH**		**DOAC**
	**Thrombocytopenia < 50 G/L**	**Thrombocytopenia ≤ 75 G/L**	**Thrombocytopenia < 50 G/L**
Continue the ongoing anticoagulant treatment	30 (8.0% [5.2–10.7])	211 (56.1% [51.1–61.1])	72 (19.1% [15.2–23.1])
Reduce the anticoagulant dosage by 50%	86 (22.9% [18.6–27.1])	18 (4.8% [2.6–6.9])	49 (13.0% [9.6–16.4])
Switch to a prophylactic dose of LMWH	46 (12.2% [8.9–15.5])	6 (1.6% [0.3–2.9])	33 (8.8% [5.9–11.6])
Switch to DOAC (if LMWH)/LMWH (if DOAC)	105 (27.9% [23.3–32.5])	85 (22.6% [18.4–26.8])	51 (13.6% [10.1–17.0])
Switch to a reduced dose of DOAC	20 (5.3% [3.0–7.6])	7 (1.9% [0.5–3.2])	39 (10.4% [7.3–13.4])
Switch to intravenous UFH	19 (5.1% [2.8–7.3])	11 (2.9% [1.2–4.6])	20 (5.3% [3.0–7.6])
Switch to Fondaparinux	70 (18.6% [14.7–22.5])	38 (10.1% [7.1–13.1])	38 (10.1% [7.1–13.1])
Continue the ongoing treatment + platelet transfusion	-	-	49 (13.0% [9.6–16.4])
Inferior vena cava filter insertion	-	-	25 (6.6% [4.1–9.2])

The values in square brackets correspond to the 95% confidence interval of the proportion. DOAC: Direct oral anticoagulant; LMWH: low molecular weight heparin; UFH: unfractionated heparin.

**Table 4 cancers-14-04143-t004:** Management of patients with VTE recurrence during anticoagulant treatment.

First-Line Management:	LMWH *	DOAC **
Increase the dosage of LMWH/ DOAC by 25%	286 (76.1% [71.7–80.4])	1 (0.3% [0.0–0.8])
Switch to a DOAC/ another DOAC	37 (9.8% [6.8–12.8])	12 (3.2% [1.4–5.0])
Switch to intravenous UFH	16 (4.3% [2.2–6.3])	5 (1.3% [0.2–2.5])
VKA bridging	3 (0.8% [0.0–1.7])	-
Switch to Fondaparinux	4 (1.1% [0.0–2.1])	2 (0.5% [0.0–1.3])
Inferior vena cava filter insertion	30 (8.0% [5.2–10.7])	12 (3.2% [1.4–5.0])
Switch to LMWH (bodyweight-adjusted dose)	-	309 (82.2% [78.3–86.0])
Switch to LMWH (125% of the bodyweight-adjusted dose)	-	35 (9.3% [6.4–12.2])

The values in square brackets correspond to the 95% confidence interval of the proportion. VTE: venous thromboembolism; DOAC: Direct oral anticoagulant; LMWH: low molecular weight heparin; UFH: unfractionated heparin; VKA: vitamin K antagonist. * Dose adjusted for bodyweight and after the ruling out heparin-induced thrombocytopenia. ** dose adjusted with marketing authorisation.

## Data Availability

Data are available by request.

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
