# Peer review of "Management of Cancer-Associated Thrombosis in France: A National Survey among Vascular Disease and Supportive Care Specialists"

_cancers, 2022, doi:10.3390/cancers14174143_

Round 1

Reviewer 1 Report

In their study by Mahé et al., the authors designed a survey for specialists regarding cancer-associated thromboembolism. The survey included several vignettes about patients with different cancer sites and questions, as indicated by the authors. Several points were taken into the consideration, such as cancer disease, anti-cancer treatments etc.

The authors found that among 376 responders, low molecular weight heparins are the first choice by most specialists. In addition, the study shows that the prescription of direct oral anticoagulants within the first 3 weeks of cancer-associated thromboembolism diagnosis was dependent on the cancer site. Although I find the study potentially interesting, it is difficult to conclude something from this study.

It would be important that the introduction and discussion include more information on LMWH, DOACs

No statistical analysis is performed, so how to consider the validity of the study?

The survey has no registration number by the authorities. Grant numbers are not indicated.

Limitations of the study are not discussed

Comparative analysis to other similar surveys could be provided in the discussion 

Author Response

REVIEWER 1

Review Report Form

Open Review

In their study by Mahé et al., the authors designed a survey for specialists regarding cancer-associated thromboembolism. The survey included several vignettes about patients with different cancer sites and questions, as indicated by the authors. Several points were taken into the consideration, such as cancer disease, anti-cancer treatments etc.

The authors found that among 376 responders, low molecular weight heparins are the first choice by most specialists. In addition, the study shows that the prescription of direct oral anticoagulants within the first 3 weeks of cancer-associated thromboembolism diagnosis was dependent on the cancer site. Although I find the study potentially interesting, it is difficult to conclude something from this study.

Reply :

We designed a survey in order to describe the prescription determinants of practitioners when facing a patient diagnosed with a CAT, in the context of new data issued from RCT with DOACs. Previously, LMWHs was the only option for treating CAT

Our conclusion is that in daily practice, LMWHs and DOACs are now considered by specialists of CAT, and that the site of cancer plays an important role in the decision.

The conclusion has been revised and key messages clarified (L43 and L387)

It would be important that the introduction and discussion include more information on LMWH, DOACs

Reply

More information on LMWHs and DOACS have been provided in the introduction (L59-65) and in the discussion (L316-320)

No statistical analysis is performed, so how to consider the validity of the study?

Reply

We confirm that there a statistical analysis was performed (section Data Analysis L 120). It’s adapted for the type of study ie a survey with a descriptive objective.

To assess the validity of the results and the study, IC 95% have been added (section results L147-228).

The survey has no registration number by the authorities.

Reply

The survey has been disseminated without any identification of asked physicians : the specialists networks proceeded by their own to the diffusion. In addition no personal data have been recorded (oncly the age class, the type of practice and specialty). Third, the data entered on a google form were directly used for the statistical treatment.

Overall, responders are totally anonymous, there is no possibility to identify responders from the recorded informations.

In this context, no specific (CNIL : Commission nationale de l'informatique et des libertés, ie French data protection authority) autorisation is required in France. This survey doesn’t come in the General Data Protection Regulation (GDPR). No ethical committee is required.

Concerning a survey, no registration number is required.

Grant numbers are not indicated.

Reply

The grant number is as follows : « CT n°FR01902036138N »

Limitations of the study are not discussed

Reply :There is a section about limitations in our manuscript (L342). This section has been implemented.

Comparative analysis to other similar surveys could be provided in the discussion 

Reply

The results of 2 recently published studies reporting surveys have been added in the discussion (L268)

-Hannah  Kaliel et al.  Retrospective review of prescribing patterns in CAT :a single center experience in Edmonton, Alerta, Canada. Clin Appl Thromb Haemost 2021 ; doi.org/10.1177%2F1076029620975489

-Maraveyas, et al. Cancer-Associated ThrOmboSIs - Patient-Reported OutcoMes With RivarOxaban (COSIMO) - Baseline characteristics and clinical outcomes . RPTH 2021 doi.org/10.1002/rth2.12604)

Reviewer 2 Report

The manuscript provides very important information from the experience of the specialists in

the field of thromboembolism. I sincerely appreciate such a manuscript and recommend it for

the publication. However, I would like to suggest minor revision due to my question

regarding the following:

Can the authors add the information about the following clinical issues that can also be

evaluated:

- comorbities like:

- presence of severe coagulopathy (massive haematomas)

- need of neuraxial catheter or lumbar puncture

- surgical intervention on spine cord / vertebral column

- allergic reaction to UFH, LMWH or DOAC

- HIT

- severe peripheral arterial disease (critical limb ischemia)

- nausea and vomiting or upper GIT intervention limiting the peroral intake of the drugs

- peripheral neuropathy

- patient´s parameters, like:

- weight of the patients, BMI

- haemoglobin level (erythropoietin treatment)

- leukocyte count

- some of the risk factors associated with the patient or with the treatment:

- hospitalization lasting for more than 6 days

- immobilization for more than 3 days

- past history of venous thromboembolism

- risk of drug-drug interactions

- severe renal failure

- increased liver tests

- atrial fibrillation

Author Response

(The authors gave the same response as above.)

Round 2

Reviewer 1 Report

The authors replied to my concers. I think that the study is suitable for the publication.